# Molecular Contrastive Pretraining with Collaborative Featurizations

## Abstract

Molecular pretraining, which learns molecular representations over massive unlabeled data, has become a prominent paradigm to solve a variety of tasks in computational chemistry and drug discovery. Recently, prosperous progress has been made in molecular pretraining with different molecular featurizations, including 1D SMILES strings, 2D graphs, and 3D geometries. However, the role of molecular featurizations with their corresponding neural architectures in molecular pretraining remains largely unexamined. In this paper, through two case studies—chirality classification and aromatic ring counting—we first demonstrate that different featurization techniques convey chemical information differently. In light of this observation, we propose a simple and effective MOlecular pretraining framework with COllaborative featurizations (MOCO). MOCO comprehensively leverages multiple featurizations that complement each other and outperforms existing state-of-the-art models that solely relies on one or two featurizations on a wide range of molecular property prediction tasks.

## 1 Introduction

Molecular representation learning, which automates the process of feature learning for molecules, is fast driving the development of computational chemistry and drug discovery. It has been recognized as crucial for a variety of downstream tasks, spanning from molecular property prediction to molecule design (Yang et al., 2019; Du et al., 2022). Deep neural models, on the other hand, rely on a substantial amount of labeled data, which require expensive wet lab experiments in chemical domains. With insufficient annotated data, deep models easily overfit to such small training data and tend to learn spurious correlations (Sagawa et al., 2020).

In recent years, self-supervised pretraining has emerged as a promising strategy to alleviate the label scarcity problem and improve model robustness (Jing & Tian, 2021). A typical framework pretrains the encoder model with training objectives over large-scale unlabeled datasets and then fine-tunes the learned model on labeled downstream tasks. Motivated by its success, many molecular pretraining models have been developed (Wang et al., 2019; Chithrananda et al., 2020; Hu et al., 2020b; You et al., 2020a; Xu et al., 2021a; Fang et al., 2022; Stärk et al., 2021; Liu et al., 2022a). To capture chemical semantics of molecules, these models design several pretraining strategies based on different *molecular featurizations*, which translate chemical information into representations that can be recognized by machine learning algorithms. For example, early models (Wang et al., 2019; Chithrananda et al., 2020) propose to leverage masked language modeling (Bengio et al., 2003) to pretrain Simplified Molecular-Input Line-Entry System (SMILES) strings (Weininger, 1988), while others study contrastive learning on 2D graphs (Hu et al., 2020b; You et al., 2020a; Xu et al., 2021a) or 3D conformations (Fang et al., 2022). Some recent studies further propose to enrich 2D-topology-based pretraining with 3D geometry information (Stärk et al., 2021; Liu et al., 2022a).

Despite encouraging progress, prior studies tend to emphasize on pretraining on molecular graphs and overlook the impact of other molecular featurizations with their corresponding neural encoders, which represent chemical information in different ways. Consider SMILES strings as an example. It explicitly represents informative structures in special characters such as branches, rings, and chirality (Ross et al., 2022), which are difficult to learn in graph-based representations (Chen et al., 2020b). Moreover, the utility of different featurizations may vary across downstream tasks. Therefore, most previous models relying on

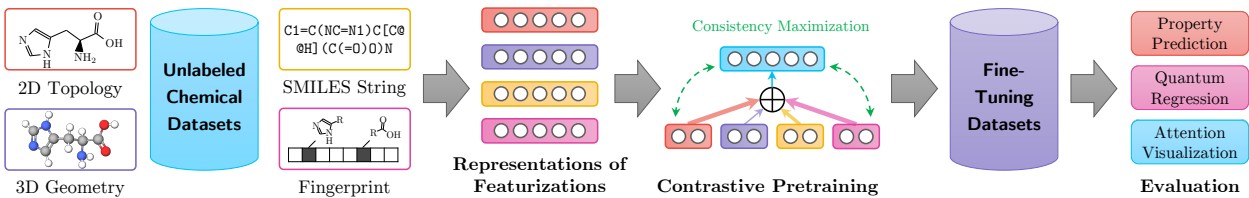

Figure 1: The proposed MOCO model. MOCO obtains four molecule featurizations with appropriate encoders. After that, an attention network is employed to aggregate each view embedding and compute a final embedding. The model is trained using a contrastive objective that maximizes the consistency between view embeddings and the final embedding.

only one or two featurizations might achieve sub-optimal performance across various downstream tasks. For example, 2D topology is important for many drug-related properties such as toxicity, while 3D geometry arguably determines properties related to quantum mechanics, such as single-point energy, atomic forces, or dipole moments (Zhang et al., 2018; Smith et al., 2017). Therefore, it is natural to ask whether we can enjoy the benefits from multiple molecular featurizations and take the relative utilities of different featurizations into consideration during fine-tuning on downstream tasks.

In this work, we first revisit four commonly used featurizations techniques: (a) 2D topology graphs, (b) 3D geometry graphs, (c) Morgan fingerprints, and (d) SMILES strings. We leverage four accompanying neural encoders with proper inductive bias and conduct two case studies, classifying tetrahedral chiral centers and counting aromatic rings, both of which are informative chemical descriptors, on representations obtained on different featurization techniques. The results show there is no one single featurization that dominates the others, indicating that different featurizations encode chemical semantics of molecules in different ways.

In light of this observation, we then propose a simple and effective MOlecular pretraining framework with COllaborative featurizations to comprehensively leverage every featurization during both pretraining and fine-tuning, which we term MOCO for brevity. Its graphical illustration is shown in Figure 1. The core idea of MOCO is to dynamically adjust the contribution of each featurization through an attention network, which *selectively* extracts information from each collaborative "view" of the raw molecular data. Besides, we design a novel multiview contrastive pretraining strategy, which trains the model by maximizing the consistency among different views in a self-supervised manner. Contrary to previous studies (Stärk et al., 2021; Liu et al., 2022a) that only consider 2D graph structures during fine-tuning, our MOCO utilizes multiple featurizations in *both* pretraining and fine-tuning stages and further allows interpretation analysis of different downstream tasks for domain scientists. Note that our proposed MOCO framework is generic, allowing for seamless integration of off-the-shelf neural architectures. To the best of our knowledge, this is the first work that studies how various featurization techniques should be utilized for molecular pretraining and downstream tasks.

We evaluate the effectiveness of our MOCO model on widely-used benchmark datasets including MoleculeNet (Wu et al., 2018) and QM9 (Ramakrishnan et al., 2014) that cover a wide range of molecular property prediction tasks. The results reveal that MOCO consistently improves non-pretraining baselines without negative transfer and outperforms existing state-of-the-art molecular pretraining models, achieving a 1.1% absolute improvement in terms of average ROC-AUC. Furthermore, the learned model weights of molecular featurizations for different end tasks are well aligned with prior chemical knowledge. We also suggest a series of guidelines on choosing effective featurization techniques for molecular representations.

The main contributions of this work are three-fold:

- We explore the featurization spaces of molecules with appropriate neural encoders and highlight the importance of incorporating different featurizations for molecular pretraining.

- We propose a novel molecular contrastive pretraining framework that adaptively integrates information from multiple collaborative featurizations during both pretraining and fine-tuning stages and provides interpretability for downstream molecular property prediction tasks.

- Extensive experiments conducted on public benchmark datasets validate the effectiveness of our proposed model. MOCO is able to achieve the state-of-the-art across various downstream datasets without negative transfer.

## 2 Preliminaries

### 2.1 A Brief Recapitulation of Molecular Featurization Techniques

Molecular featurizations translate chemical information of molecules into representations that can be understood by machine learning algorithms. Concretely, we consider the following molecular featurizations covering string-, graph-, scalar-, and vector-based representations for 1D/2D molecules and 3D structures, which are popular in literature (Ramsundar et al., 2019; Atz et al., 2021):

- **2D topology graphs** model atoms and bonds as nodes and edges respectively. It is arguably a common technique, especially for capturing substructure information by means of graph topology.

- **3D geometry graphs** incorporate atomic coordinates (conformations) in their representations and are able to depict how atoms are positioned relative to each other in the 3D space. We consider conformers in an equilibrium state, corresponding to the minima in a potential energy surface.

- **Morgan fingerprints** (Morgan, 1965; Glem et al., 2006) encode molecules in fixed-length binary strings, with bits indicating presence or absence of specific substructures. They represent each atom according to a set of atomic invariants and iteratively update these features among neighboring atoms using a hash function.

- **SMILES strings** are a concise technique that represents chemical structures in a linear notation using ASCII characters, with explicitly depicting information about atoms, bonds, rings, connectivity, aromaticity, and stereochemistry.

### 2.2 Learning Representations with Different Featurizations

Next, we introduce four encoders with different inductive bias to capture the intrinsic information with each featurization. Here we only discuss the high-level design of each encoder; please refer to Appendix A for detailed implementations of each encoder.

**Notations.** Each molecule can be represented as an undirected graph, where nodes are atoms and edges describe inter-atomic bonds. Formally, each graph is denoted as $\mathcal{G} = (\boldsymbol{A}, \boldsymbol{R}, \boldsymbol{X}, \mathsf{E})$, where $\boldsymbol{A} \in \{0, 1\}^{N \times N}$ is the adjacency matrix of $N$ nodes, $\boldsymbol{R} \in \mathbb{R}^{N \times 3}$ is the 3D position matrix, $\boldsymbol{X} \in \mathbb{R}^{N \times K}$ is the matrix of atom attributes of $K$ dimension, and $\mathsf{E} \in \mathbb{R}^{N \times N \times E}$ is the tensor for bond attributes of $E$ dimension. Additionally, each molecule is attached with a binary fingerprint vector $\boldsymbol{f} \in \{0, 1\}^F$ of length $F$ and a SMILES string $\mathbf{S} = [s_j]_{j=1}^S$ of length $S$. In what follows, the subscript $i$ is used to index the $i$-th molecule.

**Embedding 2D graphs.** To capture the 2D topological information, we employ a widely-used Graph Isomorphism Network (GIN) model (Xu et al., 2019) denoted by $f_{2D}$, which receives as input the graph adjacency matrix and attributes of atoms and bonds, and produces the embedding vector $\boldsymbol{z}_i^{2D} \in \mathbb{R}^D$:

$$\boldsymbol{z}_i^{2D} = f_{2D}(\boldsymbol{X}_i, \mathsf{E}_i, \boldsymbol{A}_i). \tag{1}$$

**Embedding 3D graphs.** To model additional spatial coordinates associated with atoms, we leverage SchNet (Schütt et al., 2017) as the backbone, which models message passing as continuous-filter convolutions and is able to preserve rotational invariance for energy predictions. We denote its encoding function as $f_{3D}$ which takes atom features and positions as input and produces the 3D embedding $\boldsymbol{z}_i^{3D} \in \mathbb{R}^D$:

$$\boldsymbol{z}_i^{3D} = f_{3D}(\boldsymbol{X}_i, \boldsymbol{R}_i). \tag{2}$$

Table 1: Results of two case studies with different featurizations: chirality classification and aromatic ring count regression.

| Target | 2D | 3D | SM | FP |
|---|---|---|---|---|
| Chirality (AP, ↑) | 0.4952 | 0.4959 | **0.5505** | 0.5246 |
| #Rings (MAE, ↓) | **0.1949** | 0.2021 | 0.3077 | 0.2590 |

**Embedding molecular fingerprints.** Since there is a lack of proper neural encoders for fingerprints, we propose an attention-based network to model interactions of feature fields in fingerprint vectors, which considers the discrete and extremely sparse nature of fingerprints. Specifically, we first transform all $F$ feature fields into a dense embedding matrix $\boldsymbol{F}_i \in \mathbb{R}^{F \times D_{\mathrm{F}}}$ via embedding lookup. Then, we use a multihead self-attention network $f_{\mathrm{FP}}$ (Vaswani et al., 2017) to model the interaction among those feature fields, resulting in an embedding matrix $\widehat{\boldsymbol{Z}}_i^{\mathrm{FP}} \in \mathbb{R}^{F \times D_{\mathrm{F}}}$. Following that, we perform sum pooling and use a linear model $f_{\mathrm{LIN}}$ to obtain the final fingerprint embedding $z_i^{\mathrm{FP}} \in \mathbb{R}^D$:

$$\widehat{\boldsymbol{Z}}_i^{\mathrm{FP}} = f_{\mathrm{FP}}(\boldsymbol{F}_i), \qquad \boldsymbol{z}_i^{\mathrm{FP}} = f_{\mathrm{LIN}}\left(\sum_{d=1}^{D_{\mathrm{F}}} \widehat{\boldsymbol{Z}}_{i,d}^{\mathrm{FP}}\right). \tag{3}$$

**Embedding SMILES strings.** To encode SMILES strings, we use a pretrained RoBERTa (Liu et al., 2019b) as the backbone model. As SMILES strings do not possess consecutive relationships, the RoBERTa model is pretrained using the masked language model as the only objective, unlike conventional natural language models (Devlin et al., 2019). After that, in order to reduce the computational burden, we freeze the RoBERTa encoder (denoted by $f_{\mathrm{SM}}$) in our model and employ an additional learnable MultiLayer Perceptron (MLP) on the representation $\boldsymbol{s}_i \in \mathbb{R}^{D_{\mathrm{S}}}$ to get the final embedding $\boldsymbol{z}_i^{\mathrm{SM}} \in \mathbb{R}^D$:

$$\boldsymbol{s}_i = f_{\mathrm{SM}}(\mathbf{S}_i), \qquad \boldsymbol{z}_i^{\mathrm{SM}} = f_{\mathrm{MLP}}(\boldsymbol{s}_i). \tag{4}$$

## 2.3 Case Studies

In this section, we present two case studies—chirality classification and aromatic ring counting—to demonstrate that the representation ability of each featurization with the corresponding neural encoder is different. For chirality classification, we randomly select 10K molecules with one chirality center from GEOM-Drugs (Axelrod & Gómez-Bombarelli, 2022) and test whether the representations obtained using the four featurizations can classify tetrahedral chiral centers as R/S. For aromatic ring counting, we randomly draw another 10K molecules and test whether these models can recognize the number of aromatic rings of each molecule. Note that both chirality properties and ring counts are informative chemical descriptors (Ritchie & Macdonald, 2009) and can be easily computed with existing implementations such as RDKit (Landrum et al., 2022).

Figure 2: (a) Chirality: even if two graphs are isomorphic, they can have two distinct stereochemistry structures. (b) The aromatic ring is an important functional group.

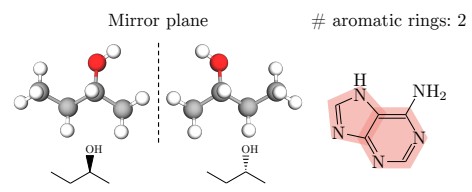

(a) $(R)$-$(-)$-2-Butanol, $(S)$-$(+)$-2-Butanol    (b) Adenine

We report classification and regression performance in Average Precision (AP) and Mean Absolute Error (MAE) respectively. The results are summarized in Table 1. It is seen from the table that no single featurization performs the best on all targets and four representations contain collaborative information to each other, suggesting us to leverage multiple featurizations for molecular pretraining.

## 3 Molecular Pretraining with Collaborative Featurizations

As with generic self-supervised learning pipelines, the MOCO framework is divided into two stages, pretraining and fine-tuning. In the first stage, given an unlabeled dataset, we train an encoding function that learns

representations with the four featurization techniques. In the subsequent fine-tuning phase, we take the weights of the encoders from the pretrained model and tune the model on molecules with annotations of particular properties in a supervised fashion.

We next introduce the MOCO pretraining framework in detail. We first use obtain four "view" representations based on the aforementioned four featurizations. Then, we integrate these four embeddings to compute a final representation for each molecule through an attention network. Finally, we pretrain the whole model using a contrastive objective.

### 3.1 Representation Aggregation from Multiple Featurizations

Since each featurization technique reflects the molecule from one certain aspect, we take weighted average of every view embedding to obtain a comprehensive final representation:

$$z_i = \sum_{m \in \mathcal{M}} \alpha^m z_i^m, \tag{5}$$

where $\mathcal{M} = \{2D, 3D, FP, SM\}$ is the set of all views. We leverage an attention network (Bahdanau et al., 2015) that learns to adjust the contribution of each view. Formally, the attention coefficient $\alpha^m$ denoting the contribution of the $m$-th view is computed by:

$$\alpha^m = \frac{\exp(w^m)}{\sum_{m' \in \mathcal{M}} \exp(w^{m'})}, \qquad w^m = \frac{1}{|\mathcal{B}|} \sum_{i \in \mathcal{B}} q^\top \cdot \tanh\left(W \frac{z_i^m}{\|z_i^m\|_2} + b\right), \tag{6}$$

where $q, b \in \mathbb{R}^D$, $W \in \mathbb{R}^{D \times D}$ are trainable parameters in the attention network, and $\mathcal{B}$ denotes the set of molecules in the current training batch. Note that we perform $\ell_2$ normalization on all embeddings to regularize the scale across different views when computing the attention scores.

### 3.2 Contrastive Objectives for Pretraining

Finally, we train the model using a contrastive objective by aligning the aggregated embedding with all view-specific embeddings. Particularly, for one molecule $i$, we designate its four view embeddings $z_i^m$ as the anchors and the aggregated embeddings $z_i$ as the positive instance. Other aggregated embeddings $\{z_j\}_{i \neq j}$ in the same batch are then chosen as the negative samples. Following prior studies (Chen et al., 2020a; He et al., 2020; Bachman et al., 2019; Zhu et al., 2020; You et al., 2020a; Zhu et al., 2021a), we leverage the Information Noice Contrastive Estimation (InfoNCE) objective, which can be formally written as:

$$\mathcal{L} = \frac{1}{|\mathcal{B}|} \sum_{i \in \mathcal{B}} \left[ \frac{1}{|\mathcal{M}|} \sum_{m \in \mathcal{M}} -\log \frac{\exp(\theta(z_i^m, z_i)/\tau)}{\sum_{j \in \mathcal{B}} \exp(\theta(z_i^m, z_j)/\tau)} \right], \tag{7}$$

where the critic function $\theta$ computes the likelihood scores of contrastive pairs and the hyperparameter $\tau$ adjusts the dynamic range of the likelihood scores of contrastive pairs. Specifically, the critic function $\theta$ performs non-linear transformation via an MLP function $g$ (Chen et al., 2020a) and then measures their cosine similarity:

$$\theta(x, y) = \frac{g(x)^\top g(y)}{\|g(x)\|_2 \|g(y)\|_2}. \tag{8}$$

After pretraining the model with the self-supervised objective function $\mathcal{L}$, we fine-tune the model weights of view encoders along with the attentive representation aggregation module with the supervision of downstream tasks at a smaller learning rate.

## 4 Experiments

In this section, we present empirical evaluation of our proposed work. Specifically, the experiments aim to investigate the following three key questions.

- **RQ1 (Overall performance).** Is the proposed MOCO able to improve non-pretraining baselines and outperform state-of-the-arts on molecular property prediction tasks?

- **RQ2 (Interpretation).** Are the learned attention weights of molecular featurizations on different downstream tasks consistent with chemical knowledge?

- **RQ3 (Ablation studies).** How do the representation aggregation module and the fine-tuning strategy affect the model performance?

In the following, we first summarize experimental setup and proceed to results and analysis.

### 4.1 Experimental Configurations

**Datasets.** We closely follow the experimental setup of GraphMVP (Liu et al., 2022a) for fair comparison. Specifically, we pretrain the model using the GEOM-Drugs dataset (Axelrod & Gómez-Bombarelli, 2022) containing both 2D and 3D information. For fine-tuning, we choose a variety datasets extracted from MoleculeNet (Wu et al., 2018), ChEMBL (Gaulton et al., 2011), and CEP (Hachmann et al., 2011), that cover a wide range of applications, including physiological, biological, and pharmaceutical tasks, and QM9 (Ramakrishnan et al., 2014) that focuses on quantum property prediction. These downstream tasks include 8 binary classification and 12 regression tasks. For those datasets for fine-tuning, we follow OGB (Hu et al., 2020a) that uses scaffolds to split training/test/validation subsets with a split ratio of 80%/10%/10%. For detailed description, we refer readers of interest to Appendix B.

**Baselines.** For comprehensive comparison, we select the following two groups of SSL methods as primary baselines in our experiments.

- Generic graph SSL models: GraphSAGE (Hamilton et al., 2017), InfoGraph (Sun et al., 2020a), GPT-GNN (Hu et al., 2020c), AttrMask, ContextPred (Hu et al., 2020b), GraphLoG (Xu et al., 2021a), GraphCL (You et al., 2020a), JOAO (You et al., 2021), and GraphMAE (Hou et al., 2022).

- Molecular SSL models: GROVER-Contextual (GROVER-C), GROVER-Motif (GROVER-M) (Rong et al., 2020), and GraphMVP[1] (Liu et al., 2022a).

In the pretraining stage, all the above SSL approaches are trained on the same dataset based on GEOM-Drugs. We also report performance with a randomly initialized model as the non-pretraining baseline. To ensure the performance is comparable with existing work, we report all baseline performance from previously published results (Liu et al., 2022a; Hou et al., 2022).

**Implementation details.** In the GEOM-Drugs dataset, since the original full set is too large (containing 317K molecules with over 9M conformations), we randomly select 50K molecules as the pretraining dataset. For each molecule, we select to use its top-5 conformers of the lowest energy in virtue of their sufficient geometry information. Since molecules in the fine-tuning datasets do not have 3D information available, we use ETKDG (Riniker & Landrum, 2015) in RDkit (Landrum et al., 2022) to compute molecular conformations. For both pretraining and fine-tuning datasets, we use RDkit to generate 1024-bit molecular fingerprints with radius $R = 2$, which is roughly equivalent to the ECFP4 scheme (Rogers & Hahn, 2010). We would like to emphasis that all dataset preprocessing and graph encoder architectures are kept in line with GraphMVP (Liu et al., 2022a) to ensure fair comparison. Readers of interest may refer to Appendix C for implementation details regarding software/hardware platforms, model training, and hyperparameter specifications.

**Evaluation protocols.** For classification tasks, we report the performance in terms of the Area Under the ROC-Curve (ROC-AUC), where higher values indicate better performance. For quantum property and other non-quantum regression tasks, we measure the performance in Mean Absolute Error (MAE) and Root Mean

---

[1]In our experiments, we do not include its two variants GraphMVP-G and GraphMVP-C since they are essentially two ensemble models that combine AttrMask and ContextPred (Hu et al., 2020b) respectively.

Table 2: Results for eight molecule property prediction tasks in terms of ROC-AUC (%, ↑). We highlight the best- and the second-best performing results in **boldface** and underlined, respectively.

| Pretraining | BBBP | Tox21 | ToxCast | SIDER | ClinTox | MUV | HIV | BACE | Avg. |
|---|---|---|---|---|---|---|---|---|---|
| — | $71.0_{\pm 0.5}$ | $\underline{75.9_{\pm 0.3}}$ | $\underline{64.7_{\pm 2.3}}$ | $57.7_{\pm 3.1}$ | $71.5_{\pm 5.3}$ | $\underline{77.7_{\pm 1.0}}$ | $75.9_{\pm 0.7}$ | $71.5_{\pm 2.7}$ | 70.63 |
| GraphSAGE | $64.5_{\pm 3.1}$ | $74.5_{\pm 0.4}$ | $60.8_{\pm 0.5}$ | $56.7_{\pm 0.1}$ | $55.8_{\pm 6.2}$ | $73.3_{\pm 1.6}$ | $75.1_{\pm 0.8}$ | $64.6_{\pm 4.7}$ | 65.64 |
| AttrMask | $70.2_{\pm 0.5}$ | $74.2_{\pm 0.8}$ | $62.5_{\pm 0.4}$ | $60.4_{\pm 0.6}$ | $68.6_{\pm 9.6}$ | $73.9_{\pm 1.3}$ | $74.3_{\pm 1.3}$ | $77.2_{\pm 1.4}$ | 70.16 |
| GPT-GNN | $64.5_{\pm 1.1}$ | $75.3_{\pm 0.5}$ | $62.2_{\pm 0.1}$ | $57.5_{\pm 4.2}$ | $57.8_{\pm 3.1}$ | $76.1_{\pm 2.3}$ | $75.1_{\pm 0.2}$ | $77.6_{\pm 0.5}$ | 68.27 |
| InfoGraph | $69.2_{\pm 0.8}$ | $73.0_{\pm 0.7}$ | $62.0_{\pm 0.3}$ | $59.2_{\pm 0.2}$ | $75.1_{\pm 5.0}$ | $74.0_{\pm 1.5}$ | $74.5_{\pm 1.8}$ | $73.9_{\pm 2.5}$ | 70.10 |
| ContextPred | $\underline{71.2_{\pm 0.9}}$ | $73.3_{\pm 0.5}$ | $62.8_{\pm 0.3}$ | $59.3_{\pm 1.4}$ | $73.7_{\pm 4.0}$ | $72.5_{\pm 2.2}$ | $75.8_{\pm 1.1}$ | $78.6_{\pm 1.4}$ | 70.89 |
| GraphLoG | $67.8_{\pm 1.7}$ | $73.0_{\pm 0.3}$ | $62.2_{\pm 0.4}$ | $57.4_{\pm 2.3}$ | $62.0_{\pm 1.8}$ | $73.1_{\pm 1.7}$ | $73.4_{\pm 0.6}$ | $78.8_{\pm 0.7}$ | 68.47 |
| GROVER-C | $70.3_{\pm 1.6}$ | $75.2_{\pm 0.3}$ | $62.6_{\pm 0.3}$ | $58.4_{\pm 0.6}$ | $59.9_{\pm 8.2}$ | $72.3_{\pm 0.9}$ | $75.9_{\pm 0.9}$ | $79.2_{\pm 0.3}$ | 69.21 |
| GROVER-M | $66.4_{\pm 3.4}$ | $73.2_{\pm 0.8}$ | $62.6_{\pm 0.5}$ | $60.6_{\pm 1.1}$ | $77.8_{\pm 2.0}$ | $73.3_{\pm 2.0}$ | $73.8_{\pm 1.4}$ | $73.4_{\pm 4.0}$ | 70.14 |
| GraphCL | $67.5_{\pm 3.3}$ | $75.0_{\pm 0.3}$ | $62.8_{\pm 0.2}$ | $60.1_{\pm 1.3}$ | $78.9_{\pm 4.2}$ | $77.1_{\pm 1.0}$ | $75.0_{\pm 0.4}$ | $68.7_{\pm 7.8}$ | 70.64 |
| JOAO | $66.0_{\pm 0.6}$ | $74.4_{\pm 0.7}$ | $62.7_{\pm 0.6}$ | $60.7_{\pm 1.0}$ | $66.3_{\pm 3.9}$ | $77.0_{\pm 2.2}$ | $76.6_{\pm 0.5}$ | $72.9_{\pm 2.0}$ | 69.57 |
| GraphMVP | $68.5_{\pm 0.2}$ | $74.5_{\pm 0.4}$ | $62.7_{\pm 0.1}$ | $\mathbf{62.3_{\pm 1.6}}$ | $79.0_{\pm 2.5}$ | $75.0_{\pm 1.4}$ | $74.8_{\pm 1.4}$ | $76.8_{\pm 1.1}$ | 71.69 |
| GraphMAE | $70.9_{\pm 0.9}$ | $75.0_{\pm 0.4}$ | $64.1_{\pm 0.1}$ | $59.9_{\pm 0.5}$ | $\underline{81.5_{\pm 2.8}}$ | $76.9_{\pm 2.6}$ | $\underline{76.7_{\pm 0.9}}$ | $\underline{81.4_{\pm 1.4}}$ | 73.31 |
| MOCO | $\mathbf{71.6_{\pm 1.0}}$ | $\mathbf{76.7_{\pm 0.4}}$ | $\mathbf{64.9_{\pm 0.8}}$ | $\underline{61.2_{\pm 0.6}}$ | $\mathbf{81.6_{\pm 3.7}}$ | $\mathbf{78.5_{\pm 1.4}}$ | $\mathbf{78.3_{\pm 0.4}}$ | $\mathbf{82.6_{\pm 0.3}}$ | **74.41** |

Squared Error (RMSE) respectively, where lower values are better. We repeat every experiment on three seeds with scaffold splitting and report the averaged performance with standard deviation, following previous work (Liu et al., 2022a).

## 4.2 Main Results on Molecular Property Prediction

The performance of molecular property prediction tasks is summarized in Table 2. It can be found that our MOCO shows strong empirical performance across all eight low-data downstream datasets, delivering seven out of eight state-of-the-art results and acquiring a 1.1% absolute improvement on average. The outstanding results validate the superiority of our proposed model.

We make other observations as follows. Firstly, MOCO obtains more accurate and stabler predictions compared to the randomly initialized baseline, indicating that our pretraining framework can transfer the knowledge from large, unannotated datasets to smaller downstream datasets without negative transfer. Secondly, previous work has already achieved pretty high performance. For example, the current state-of-the-art GraphMVP only obtains a 0.8% absolute improvement over its best baseline ContextPred in terms of average ROC-AUC. Our work pushes that boundary without extensive hyperparameter tuning, with an absolute improvement of up to 3.4% over GraphMVP in terms of average ROC-AUC. Lastly, it is worth mentioning that, the non-pretraining baseline even achieves better performance than some graph-based pretraining models. On some challenging datasets (e.g., Tox21, MUV, and ToxCast), it even achieves the second to best performance. This once more demonstrates the effectiveness of leveraging multiple featurization techniques.

## 4.3 Interpretation and Analysis

In order to analyze the correlation between tasks and featurization techniques, we visualize the attention weights $\boldsymbol{\alpha}$ learned on different downstream tasks in Figure 3. Note that most of the datasets in MoleculeNet (Wu et al., 2018) are ADMET property prediction tasks: chemical Absorption (A), Distribution (D), Metabolism (M), Excretion (E), and Toxicity (T), and we thus group the eight end tasks according to their prediction targets in the following analysis.

In general, we can interpret from the visualization that *2D-based features are more significant than 3D-based features in the studied tasks*, which is well aligned with chemical knowledge. We provide detailed analysis as follows:

- In Tox21, ClinTox, SIDER, and ToxCast, we find that 2D graphs play the most important role. These four datasets are related to toxicity (or side effects). Although it is a very complex biological issue to explain, such properties can still be partially deduced from certain functional groups patterns contained in 2D graphs. Actually, medicinal chemists have developed such a database to provide them with necessary alerts of potential side effects in drug design (Baell & Holloway, 2010).

- BBBP, which measures blood-brain barrier permeability, is mostly dominated by the following properties: liposolubility/water-solubility, molecular weight, and interaction between molecules and transporter proteins. Similarly, these properties can also be inferred from 2D topology, such as molecules with too many hydrogen bond acceptors/donors are unlikely to break the blood-brain barrier due to poor liposolubility (Suckling et al., 1986).

- On BACE and MUV we see 2D graphs and SMILES strings contribute most. These two datasets are about predicting protein-ligand binding activities, which are theoretically relevant to 3D conformations. However, it is still an open question that whether the conformation sampling methods can produce conformations that resemble bioactive conformations, which provide the key information for protein-ligand binding. Nevertheless, in each of these tasks, the target protein is fixed so that bioactivity can be partially deduced from 2D structures, which is supported by the success of fragment-based Quantitive Structure-Activity Relationship (QSAR) models (Manoharan et al., 2010).

- Due to the complicated pathogenetic mechanisms, it is hard to draw an explanation to why attention weights of fingerprints outweigh the other three features in the HIV task. Given that the HIV dataset is the largest one (over 40,000 molecules per task), one possible explanation of this phenomenon is that we use a high-dimensional fingerprint representations (1024 bits).

Concerning the difference between three 2D-based features (namely 2D topological graphs, fingerprints, and SMILES strings), we make the following findings, which we hope could serve as guidelines for future research on molecular representation learning:

- 2D graph representations can encode local information explicitly by resembling chemical structures. Besides, graph-based neural networks can capture long-range local chemical environment through message passing. For example, with molecular graphs, it is more convenient to identify which part of the molecule serves as a scaffold.

- In principle, SMILES strings contain all 2D information of certain molecules, but with atoms and bonds represented in ASCII characters, neural networks may have difficulty in distilling semantic meanings of chemical structures in a numerical way.

- Fingerprint representations are based on local structures and thus such features may be less effective in circumstances where long-range effects induced by topologically distant functional groups predominate, which accounts for relatively small attention weights of fingerprints in Figure 3.

### 4.4 More Experiments on Molecular Property Regression

To demonstrate that the conformations generated by RDKit are helpful, we further conduct an experiment on quantum property regression on the QM9 dataset (Ramakrishnan et al., 2014), where 3D conformations generated by RDKit are used for the fine-tuning datasets. This task is known to be closely related to 3D structures. Table 3 presents the performance comparison of MEMO with two non-pretraining (supervised) baselines SchNet and MOCO (denoted by SchNet-NP and MOCO-NP) and two state-of-the-art pretraining baselines GraphMVP (Liu et al., 2022a) and 3D Infomax (Stärk et al., 2021).

It is seen that our MOCO model achieves the best performance on all datasets. GraphMVP that consider only 2D structures

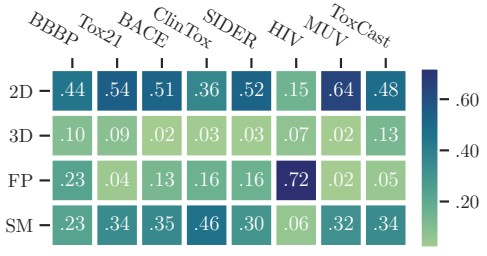

Figure 3: Visualizing the learned attention weights on eight molecular property prediction datasets.

Table 3: Results for eight molecule quantum property regression tasks in terms of Mean Absolute Error (MAE, ↓). The highest performance is highlighted in **bold**.

| Target
Unit | $\mu$
D | $\alpha$
Bohr$^3$ | $\epsilon_{\text{HOMO}}$
meV | $\epsilon_{\text{LUMO}}$
meV | $\epsilon_{\text{gap}}$
meV | $U_0$
meV | $U$
meV | $\langle R^2 \rangle$
Bohr$^3$ |
|---|---|---|---|---|---|---|---|---|
| SchNet-NP | 0.4604 | 0.3251 | 95.9740 | 78.5870 | 136.4720 | 98.1240 | 100.1650 | 24.3277 |
| MOCO-NP | 0.3767 | 0.2439 | 73.0625 | 69.8780 | 102.2332 | 77.4708 | 92.8562 | 17.5842 |
| GraphMVP | 0.3726 | 0.4390 | 75.3750 | 72.3820 | 104.8370 | 278.8900 | 325.8021 | 22.6433 |
| 3D Infomax | 0.3644 | 0.4190 | 72.0558 | 67.6203 | 99.4032 | 207.2148 | 219.5415 | 20.3934 |
| MOCO | **0.3618** | **0.2236** | **71.5120** | **58.5890** | **97.7440** | **64.3550** | **66.3958** | **15.5571** |

during fine-tuning even result in negative transfer on some
datasets. Our MOCO, on the contrary, achieves better performance than the supervised baseline, underscoring the value of leveraging 3D structures (as well as other sources of 2D information) during fine-tuning.

We also perform experiments on non-quantum property regression tasks. Our proposed MOCO also obtains promising improvements compared to the current state-of-the-art baselines. Please refer to Appendix D.1 for performance comparison and analysis.

## 4.5 Ablation Studies

Finally, we conduct ablation studies on the representation aggregation module and the fine-tuning strategy. We consider the following model variants for further inspection. Except the modifications in specific modules, other implementations remain the same as previously described.

- **MOCO–Max** removes the attention network in the representation aggregation module in Equation (5) and simply uses max pooling to combine view embeddings.

- **MOCO–Mean** modifies representation aggregation by taking average over view embeddings.

- **MOCO–Freeze** does not fine-tune the representation aggregation module but instead uses the frozen weights of the pretrained model.

We report the performance of model variants in Figure 4. It is seen that all three variants achieve downgraded performance, which empirically rationalizes the design choice of our molecular pretraining framework with collaborative featurizations. Specifically, the performance of MOCO–Max and MOCO–Mean without attention aggregation mechanisms of multiple featurizations is inferior to that of MOCO, demonstrating the necessity of adaptively combining information from multiple featurizations. In addition, MOCO–Freeze occasionally obtains better performance than the two other variants, which indicates that our proposed attention network is

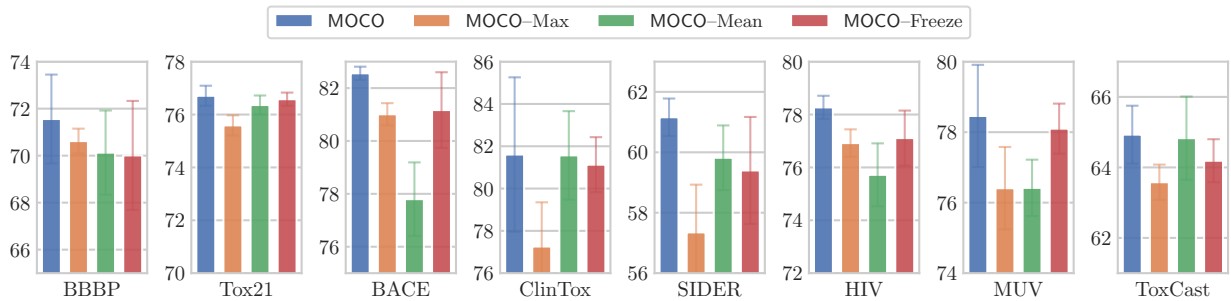

Figure 4: Ablation studies on representation aggregation and the fine-tuning strategy.

able to select information from different views. It does not, however, fine-tune the contribution of featurizations with downstream datasets, where the optimal combination might differ, resulting in performance deterioration.

Moreover, we conduct ablation studies on models that include only three view representations, where the results can be found in Appendix D.2. Results demonstrate the necessity of comprehensively leveraging four views in the proposed MOCO model.

## 5 Related Work

Traditional methods (Carhart et al., 1985; Nilakantan et al., 1987; Rogers & Hahn, 2010) represent molecular structures with fingerprints. Some prior studies (Svetnik et al., 2004; Meyer et al., 2019; Wu et al., 2018) employ tree-based machine leaning models such as random forests (Breiman, 2001) and XGBoost (Chen & Guestrin, 2016) on fingerprints to predict the properties of molecules. With the development of deep learning, neural approaches have been dominating the field given their strong representation ability. One line of work (Wang et al., 2019; Chithrananda et al., 2020) leverages language modeling techniques such as BERT (Devlin et al., 2019) to learn molecular representations based on SMILES strings (Weininger, 1988). However, some argue that sequence-based representations cannot fully capture substructure information and propose to leverage Graph Neural Networks (GNNs), which model molecules as graphs with atoms as nodes and bonds as edges (Gilmer et al., 2017; Liu et al., 2019a; Ying et al., 2021). Despite the prosperous progress, they only model 2D topological structures of molecules, without considering the 3D coordinates of atoms that are known to determine certain chemical and physical functionalities of molecules. To address this deficiency, recent work further explicitly considers such 3D geometry and designs equivariant networks to obtain the representations (Schütt et al., 2017; Klicpera et al., 2020; Satorras et al., 2021; Fuchs et al., 2020; Schütt et al., 2021; Du et al., 2021; Liu et al., 2021; Gasteiger et al., 2021; Batzner et al., 2021; Brandstetter et al., 2022; Xu et al., 2021b).

Even though molecular representation learning techniques have been extensively investigated, there are very few labeled datasets available for studying the molecular properties of interest (e.g., drug-likeness or quantum properties). On the other hand, there are abundant unannotated molecules available, which motivates researchers to study pretraining techniques that learn the model weights in a self-supervised manner and transfer the knowledge to downstream datasets with limited annotations via fine-tuning. A series of pretraining frameworks on 2D molecular graph representations have been developed so far (Rong et al., 2020; Hu et al., 2020b; Zhang et al., 2021; Wang et al., 2022; Li et al., 2020). Recent work GEM (Fang et al., 2022) studies large-scale pretraining for 3D geometry representations. Additionally, researchers also study to supplement 2D-graph-based pretraining with 3D conformation information (Yang et al., 2021; Liu et al., 2022a; Stärk et al., 2021).

A succinct comparison of our work with other representative methods is provided in Table 4. Compared to the above studies, our proposed MOCO is the only model that can *adaptively* leverage multiple featurizations for both pretraining and fine-tuning stages.

## 6 Conclusions and Discussions

This paper examines different featurizations for molecular data and highlights the importance of incorporating multiple featurizations during both pretraining and fine-tuning. Then, we develop a novel pretraining framework MOCO with collaborative featurizations for molecular data, which is able to adaptively distill information from each featurization and allows interpretability from the learned model weights. Extensive experiments on a wide range of property prediction benchmarks show that MOCO consistently outperforms existing baselines without negative transfer.

The study of featurization techniques for molecular machine learning in general remains widely open. We would like to acknowledge that the relative utility of various featurizations for different molecular predictive tasks could be usefully explored in further work. Moreover, more future research should be undertaken to specifically analyze the relationship between several featurizations, the representation ability of corresponding neural architectures, as well as the task-featurization correlation.

Table 4: Comparing MOCO with representative self-supervised methods on molecular pretraining.

| Method | Pretraining | | | | Fine-tuning | | | |
|---|---|---|---|---|---|---|---|---|
| | 2D | 3D | Fingerprint | SMILES | 2D | 3D | Fingerprint | SMILES |
| SMILES-BERT (Wang et al., 2019) | | | | ✓ | | | | ✓ |
| ChemBERTa (Chithrananda et al., 2020) | | | | ✓ | | | | ✓ |
| AttrMask, ContexPred (Hu et al., 2020b) | ✓ | | | | ✓ | | | |
| GraphCL (You et al., 2020a) | ✓ | | | | ✓ | | | |
| GraphLoG (Xu et al., 2021a) | ✓ | | | | ✓ | | | |
| GROVER (Rong et al., 2020) | ✓ | | | | ✓ | | | |
| GEM (Fang et al., 2022) | | ✓ | | | | ✓ | | |
| 3D Infomax (Stärk et al., 2021) | ✓ | ✓ | | | ✓ | | | |
| GraphMVP (Liu et al., 2022a) | ✓ | ✓ | | | ✓ | | | |
| MOCO (Ours) | ✓ | ✓ | ✓ | ✓ | ✓ | ✓ | ✓ | ✓ |

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
