# OpenReview forum: "Improving Molecular Pretraining with Collaborative Featurizations"
_TMLR — Rejected by TMLR_

### Review · Reviewer_5QhU · 2023-02-14

**Summary Of Contributions:**

In this paper, a new method for contrastive pre-training is introduced. The Authors argue that multiple featurizations of molecules can lead to better performance as each representation learns different aspects of the problem. Hence, the proposed model, MOCO, uses four molecular featurizations: 2D topology, 3D conformations, SMILES, and structural fingerprints. In the pre-training phase, the contrastive loss is used to push all the representations towards the aggregated view. The experiments show that models trained this way achieve superior results in fine-tuning for various chemical tasks. What is more, the model uses attention to select featurizations, so the importance of each featurization can be inspected after training.

**Audience:**

Yes

**Broader Impact Concerns:**

No concerns raised, and I do not think such a section would be necessary for this paper.

**Claims And Evidence:**

Yes

**Requested Changes:**

The weaknesses mentioned above should be addressed by the Authors. There are further minor changes I would like to suggest:
- Figure 2 should be referenced in the paper
- In section 4.1, implementation details, there is a typo: “We would like to emphasis…” -> “We would like to emphasize…”


**Strengths And Weaknesses:**

Strengths:
- The novelty of this paper lies in combining strong model architectures and discovering that adding fingerprints and SMILES strings can boost the performance.
- MOCO, after training, can attribute its predictions to the input featurizations (by calculating featurization importance of the attention layer).
- The motivation of the paper is very clear, and there are examples showing why different featurizations may matter.
- A good representation of featurizations is used, ranging from graph representations to vector and text representations. I am curious if using alternative featurizations (e.g. SELFIES or path-based substructural fingerprints) could lead to even better performance.
- The preliminaries section introduces different featurizations in an understandable way, and the math notation is easy to follow.
- I appreciate that the Authors spend their time and efforts to conduct case studies that support the motivation of the paper. Indeed, it seems that different featurizations excel in different tasks.
- Although improvement demonstrated in Table 2 is not huge (see the standard deviations and no statistical tests confirming the superior performance), MOCO consistently wins in 7 out of 8 tasks. The results in Table 3 are impressive, yet more models could be added to this comparison.

Weaknesses:
- The technical novelty is rather limited. The idea is similar to the one presented in the GraphMVP paper but extended to more featurizations.The used pre-training procedure is not new either. However, the selection of strong architectures that can be combined using attention pooling certainly required a lot of work.
- The Authors state that their model “provides interpretability for downstream molecular property tasks.” However, the interpretations provided by the model only explain which featurizations were used by the model. Based on that description, the readers may expect more fine-grained interpretations, but the used architectures are not interpretable. I would suggest rephrasing this contribution.
- The statement “branches, rings, and chirality [...] are difficult to learn in graph-based representations (Chen et al., 2020b)” is inaccurate in my opinion. The cited paper shows that it is difficult for GNNs to **count** these substructures, but detecting them should be no problem. Could you please provide more evidence or rephrase this statement? Moreover, in Table 1, the GNN encoder is the best among all the tested models in counting rings.
- The embedding lookup is mentioned when explaining the molecular fingerprints embedding. This does not explain how the embeddings are created. Are they trained using an embedding layer? Or by embedding lookup you mean something more than just finding embeddings by indices?
- In the experiments, GraphMVP-G and GraphMVP-C are not included. I think the decision to exclude these models from the comparison is not fair as they use equally valid pre-training methods, only slightly extended compared to GraphMVP.

---

> ### Author Response · Authors · 2023-03-20
> **Response to Reviewer 5QhU (pt. 1/2)**
>
> Thank you for your constructive feedback and encouraging comments. For your main concerns, we would like to make the following clarifications.
>
> **Q1: I am curious if using alternative featurizations (e.g. SELFIES or path-based substructural fingerprints) could lead to even better performance.**
>
> Thank you for bringing this issue to our attention.  It would indeed be interesting to see a study covering a broader range of featurizations.  We further consider two new featurization techniques published in 2020: SELFIES [1], which is based on string representations and fragprints [2], which is a path-based substructural fingerprints. We replace SMILES and fingerprints with SELFIES and fragprints respectively and keep other experimental protocols the same. The results on molecular property prediction are shown below:
>  |      | BBBP | Tox21 | Sider | Clintox | Bace | MUV | HIV | Toxcast | Avg. |
> |------|------|-------|-------|---------|------|-----|-----|---------|------|
> | Selfies | 70.6 | 76.5 | 62.0 | 79.8 | 82.1 | 77.5 | 78.4 | 64.1 | 73.9 |
> | Fragprints | 68.1 | 75.0  | 58.3  | 78.6 | 71.1 | 77.0 | 77.1 | 63.4 | 71.1 |
> | MOCO | 71.6 | 76.7 |  61.2  | 81.6 | 82.6 | 78.5 | 78.3 | 64.9 | 74.4 |
>
>
> [1] M. Krenn et al., Self-Referencing Embedded Strings (SELFIES): A 100% Robust Molecular String Representation, Mach. Learn.: Sci. Technol., 2020
>
> [2] A. Thawani et al., The Photoswitch Dataset: A Molecular Machine Learning Benchmark for the Advancement of Synthetic Chemistry, ICLR Workshop on Fundamental Science in the Era of AI, 2020
>
>
> **Q2: The technical novelty is rather limited.**
>
> Thanks for your valuable comment. Please refer to our general response regarding the novelty and the technical contribution of this work.Moreover, our work focuses on empirical analysis and the practical implications of using multiple featurizations in pretraining and fine-tuning for molecular representation learning, which we believe is a valuable contribution to the field. Furthermore, we kindly note that, as per TMLR guidelines, novelty of the method is not a necessary criterion for acceptance. ("Nor should it form the basis for rejecting work on a method considered not “novel enough”, as novelty of the studied method is not a necessary criteria for acceptance.")
>
> **Q3:  Rephrasing the contribution that their model “provides interpretability for downstream molecular property tasks”.**
>
> Thank you for your suggestions. We will rephrase our contribution to avoid the confusions. Our model provides the interpretability for promising featurizations under the given models which are widely-used and effective backbones for each featurization in the field. Therefore, we conducted corresponding interpretability study based on the models and the attention map on representations learned by each encoder. We hope to shed light on the interpretability of the task (some populations of the molecules) why certain featurization is focused more than others. However, we do agree that this is far away from anything with principle, we leave this to our future work.
>
> **Q4: Does not explain how the embeddings are created.(embedding layer or other methods)**
>
> Thank you for pointing this out. We actually elaborate the setting of fingerprint embeddings in the Appendix A, which combines the embedding lookup and positional embedding matrix to cope with discrete and extremely sparse nature of fingerprint vectors and capture the positional relationship among bits, respectively. And we will clarify how the embeddings are created in the revised version for easy understanding.

---

> ### Author Response · Authors · 2023-03-20
> **Response to Reviewer 5QhU (pt. 2/2)**
>
> **Q5: GraphMVP-G and GraphMVP-C are excluded, which is not fair to compare.**
>
> Thank you for your careful review. As mentioned in the GraphMVP, they pick the empirically optimal generative and contrastive 2D SSL method: that is AttrMask for GraphMVP-G and ContextPred for GraphMVP-C. However, these strategies are common self-supervised learning techniques which can be added to any of the baseline models and our model design. Thus, we do not include these two variants for fair comparison.
>
> **Q6: The statement “branches, rings, and chirality [...] are difficult to learn in graph-based representations (Chen et al., 2020b)” is inaccurate.**
>
> Thank you for bringing this issue to our attention. We would like to clarify that our original intention was to highlight the limitations of graph neural networks in distinguishing all molecular structures and emphasize the potential of other featurizations to compensate for these limitations. While GNNs have shown remarkable performance in various tasks, it is crucial to acknowledge that they may not be able to capture all the nuances of molecular structures. Our work aims to demonstrate that incorporating representations based on other featurizations can help overcome the limitations of GNNs and lead to a more comprehensive understanding of molecular structures. In our study, by leveraging these complementary featurizations, our approach provides a more holistic representation of molecules, thereby addressing the shortcomings of GNNs and improving the overall performance on various downstream tasks.
>
> Moreover, we will revise the omission and mistake mentioned in the requested changes. Once again, we appreciate it for your insightful suggestions which enable us to further strengthen our work. We expect that our response will resolve your main concerns.

---

### Review · Reviewer_P88d · 2023-03-03

**Summary Of Contributions:**

This work aims to design a pre-training method for learning on molecules. After reviewing necessity of four popular molecule featurization techniques, it designs an attention-based combination as collaborative feature, which is then fed into pre-training and fine-tuning. Finally, the authors provide ablation study to discuss how and why collaborative features achieve better empirical performance.

Contributions:
1. A straightforward and effective design of collaborative featurization of various molecule embeddings
2. Outstanding empirical performance across several datasets against standard baselines
3. Discussion on relative contribution from different modules for final performance: on weights of molecule embeddings, and on pre-training or not

**Audience:**

Yes

**Broader Impact Concerns:**

No.

**Claims And Evidence:**

Yes

**Requested Changes:**

As discussed above.

**Strengths And Weaknesses:**

Pros:
1. The proposed framework is straightforward but effective, with outstanding experimental performance.
2. Ablation study on different modules is sufficient, leaving some interesting questions for future work, such as 2D vs 3D vs FP vs SMILES.
3. It is well written and easy to understand.

Concerns:
1. Although in Section 4.2 the authors have mentioned ``the non-pretraining baseline even achieves better performance than some graph-based pretraining models``, could you please reveal more on why the non-pretrained is better than these pre-trained baselines? Is it related to some inherent nature or hardness of tasks, or necessity of pre-training in this kind of problems?
2. For now, the basic molecule features are combined with attention weights from separate outputs of four encoders in Section 2.2. Is it possible to combine them inside the encoders to rich their interactions? For example, on 2D graphs, additional node features with substructure info turn out to improve the performance of MPNNs by a large margin (Bouritsas el al).

Reference:
* Bouritsas et al. Improving Graph Neural Network Expressivity via Subgraph Isomorphism Counting.

---

> ### Author Response · Authors · 2023-03-20
> **Response to Reviewer P88d**
>
> Thank you for your constructive feedback and encouraging comments. For your main concerns, we would like to make the following clarifications.
>
> **Q1: Although in Section 4.2 the authors have mentioned the non-pretraining baseline even achieves better performance than some graph-based pretraining models, could you please reveal more on why the non-pretrained is better than these pre-trained baselines? Is it related to some inherent nature or hardness of tasks, or necessity of pre-training in this kind of problems?**
>
> Thank you for your careful review. It is true that our downstream model without pre-training outperforms some pre-trained models, as you mentioned, which reflects the necessity of including different featurizations  in downstream tasks. While a single featurization may contain sufficient information to be mined, existing models struggle to learn the intrinsic molecular semantics. Therefore, multiple  featurizations to some extent show the information that they respectively emphasize, which is more conducive to model learning. Additionally, although MOCO without pre-training outperforms other pre-trained models, pre-training is still effective in downstream scenarios with multiple  featurizations, as shown in the first and last rows of Table 1 in our paper.
>
> **Q2: For now, the basic molecule features are combined with attention weights from separate outputs of four encoders in Section 2.2. Is it possible to combine them inside the encoders to rich their interactions? For example, on 2D graphs, additional node features with substructure info turn out to improve the performance of MPNNs by a large margin (Bouritsas el al).**
>
> We understand that incorporating information from different featurizations directly within the encoders could potentially enrich their interactions and improve performance, as demonstrated by Bouritsas et al. for 2D graphs.
> While integrating information from various featurizations inside the encoder is a plausible approach, it relies on the assumption that explicitly encoded information is efficient and useful for the specific task. For the multiple featurizations we employed, the input data is abstract and difficult to understand, which could increase the complexity of model design and training if we were to explicitly incorporate it.
> Instead, we believe that utilizing a carefully designed encoder to mine and extract a single featurization for obtaining more representative representations, and then conducting interactions based on these representations, can provide a more comprehensive solution. This research paradigm is commonly used and effective in traditional multimodal tasks, as evidenced by Rao et al.'s DenseCLIP [1].
> We appreciate your valuable feedback and will consider exploring the possibility of combining features within encoders in future work.
>
> [1] DenseCLIP: Language-Guided Dense Prediction with Context-Aware Prompting. Rao et.al. CVPR 2022

---

### Review · Reviewer_2Ygj · 2023-03-04

**Summary Of Contributions:**

This paper proosed a framework for molecule pretraining. The main framework is to use the standard contrastive learning of molecule representations, where the NCE is deployed as a loss function. The core idea is to combine the representaion of molecules from different low-level features, including 1) 2D graph 2) 3D graph 3) SMILES string 4) molecule fingerprints, and the resulting representation is a weighted combination of each of these four representations. Experipents are done by first pre-training on a 50K molecule dataset, and then the performance on downstream tasks are evaluated and compared against existing methods, where overall 1.1% absolute improvemet is obtained.



**Audience:**

Yes

**Claims And Evidence:**

Yes

**Requested Changes:**

- I’d like to see how scalable this appraoch would be, interms of the benefit from pretraining on more data. Or at least if one can pretrain on the same dataset as most of the existing approaches, then that would have better alignment.
- I’d also like to hear the response to my above questions in the ‘weakness’ part.


**Strengths And Weaknesses:**

Strength:

- The proposed method is simple.
- The gain obtained on downstream tasks seems to be interesting.
- The paper is written in a way that is easy to understand.

Weakness:

- Despite that I like the simple approach, there is not too much technical depth or insight into this feature engineering trick for molecule representation.

- It is a bit strange to me that the authors have filtered the pretraining data to a subset of 50k molecules, which is pretty small compared to the dataset people has used in the domain. Also it may raise the concern that it is not sure if the data filtering actually helps the training, or at least it is hard to compare against other methods if pre-trained on different datasets.

- As the main contributoin of the work is the feature engineering part, it would make more sense to focus on this part, in terems of: 1) whether it would be helpful for other pretraining objectives, or it is just useful for contrastive learning; 2) whether the feature combination itself can already be helpful than using individual ones.  I guess the answers for both questions should be true, but I’d like to see justifications.

---

> ### Author Response · Authors · 2023-03-20
> **Response to Reviewer 2Ygj**
>
> Thank you for your helpful feedback. Please see our response to your concerns.
>
> **Q1：Despite that I like the simple approach, there is not too much technical depth or insight into this feature engineering trick for molecule representation.**
>
> Thanks for your valuable comment. Please refer to our general response regarding the novelty and the technical contribution of this work.
>
> **Q2: It is a bit strange to me that the authors have filtered the pretraining data to a subset of 50k molecules, which is pretty small compared to the dataset people has used in the domain. Also it may raise the concern that it is not sure if the data filtering actually helps the training, or at least it is hard to compare against other methods if pre-trained on different datasets.**
>
> Thank you for your careful review. We chose 50K data for pretraining mainly for two reasons. Firstly, to ensure a fair comparison, we strictly followed the experimental settings of GraphMVP for pretraining. It is worth noting that our data selection method is random, rather than based on specific criteria, which would not lead to distribution shift. Moreover, all baseline methods were pre-trained using the same size and completely identical data and then fine-tuned, so we have maximally ensured the fairness of the comparison.
> Secondly, whether different pretraining data sizes are really effective in molecular pretraining research is still an issue that needs to be explored in depth. GraphMVP claim that “with 50K pretraining data scale, generally we can match with the original paper, even though most of them are using larger pre-training datasets, like ZINC-2m”. Moreover, previous study [2] has shown that increasing the data size does not bring significant gains, and they give several reasons of why the self-supervised pretraining may not be very effective. However, while ensuring fairness of comparison, we used a smaller pretraining data size to save computational costs and achieved promising results.
>
> We also conducted additional pre-training on a larger scale, and the results are shown in the table below.
>
> |         | BBBP | Tox21 | Sider | ClinTox | Bace | MUV  | HIV  | Toxcast | Avg.  |
> |---------|------|-------|-------|---------|------|------|------|---------|-------|
> | MOCO_50K| 71.6 | 76.7  | 61.2  | 81.6    | 82.6 | 78.5 | 78.3 | 64.9    | 74.43 |
> | MOCO_100K| 70.4| 76.2  | 62.6  | 83.7    | 80.0 | 80.4 | 77.6 | 64.9    | 74.46 |
>
>
> [2] Does GNN Pretraining Help Molecular Representation? Sun et.al. NeurIPS 2022
>
> **Q3: As the main contribution of the work is the feature engineering part, it would make more sense to focus on this part, in terms of: 1) whether it would be helpful for other pretraining objectives, or it is just useful for contrastive learning; 2) whether the feature combination itself can already be helpful than using individual ones. I guess the answers for both questions should be true, but I’d like to see justifications.**
>
> 1）Thank you for your comment regarding the main contribution of our work being the feature engineering aspect. We would like to emphasize that contrastive learning has become increasingly popular and has demonstrated impressive empirical performance across a wide range of tasks. This motivated our choice to focus on contrastive learning in the current study.
> Secondly, it is worth investigating the applicability of our feature engineering approach to other pretraining objectives. In our work, we have considered other generative models as baselines, such as AttrMask and ContextPred in the case of 2D graphs and SMILES-BERT for 1D SMILES strings. However, we believe that designing an integrated generative pretraining objective for different featurizations might be challenging. As such, we consider this a future direction to explore.
>
> 2）We have included the ablation study of removing one featurization in Appendix D.2. The results show that removing any one featurization leads to a decrease in performance, demonstrating the importance of the four featurization fusion. We further conduct ablation study of keeping one featurization and the results are shown in the following:
>
> |            | BBBP | Tox21 | Sider | ClinTox | Bace | MUV  | HIV  | Toxcast | Avg.  |
> |------------|------|-------|-------|---------|------|------|------|---------|-------|
> | SMILES      | 70.7 | 76.0  | 58.7  | 81.0    | 81.6 | 73.2 | 73.5 | 64.4    | 72.4  |
> | 2D         | 65.4 | 74.9  | 58.0  | 58.8    | 72.6 | 71.0 | 75.3 | 61.6    | 67.2  |
> | 3D         | 61.7 | 74.0  | 59.6  | 69.9    | 81.7 | 73.7 | 67.9 | 64.4    | 69.1  |
> | Fingerprints | 66.3 | 73.4  | 51.0  | 64.4    | 76.3 | 76.9 | 70.5 | 61.1    | 67.5  |
> | MOCO       | 71.6 | 76.7  | 61.2  | 81.6    | 82.6 | 78.5 | 78.3 | 64.9    | 74.4  |
>
>
> It is seen from the table that our MOCO model achieves the best performance for all of the datasets, which demonstrates the effectiveness of our proposed multiview fusion module.

---

### Review · Reviewer_k4tC · 2023-03-09

**Summary Of Contributions:**

This work focuses on learning high-quality molecular representations for molecular property prediction. It points out that different molecule featurizations, or equivalently different ways to encode molecules in digital data, have their distinct advantages for various downstream molecular property prediction tasks. Motivated by this finding, the authors propose a MOlecular pretraining framework with COllaborative featurizations (MOCO) that leverage multiple featurizations during both pretraining and fine-tuning stages, in order to integrate the advantages from each featurization and achieve the optimal performance for a variety of downstream tasks. The authors present experimental results on multiple public benchmark datasets and claim the state-of-the-art performances.

**Audience:**

Yes

**Claims And Evidence:**

Yes

**Requested Changes:**

Please address the weaknesses above with sufficient discussions and clear explanations.

**Strengths And Weaknesses:**

Strengths
+ The presentation is easy to follow.
+ Section 2.3 is a solid motivation for the proposed method.
+ The ablation studies are well designed, although it would be better to show some results (e.g. Appendix D2) in main text.

Weaknesses
- The novelty is limited. There are recent works that also leverage multiple featurizations [1].
- The reported numbers, e.g. Table 2, are lower than simple supervised baselines reported in [2]. And some metrics do not follow the recommendations in [2] (PRC-AUC vs ROC-AUC). In addition, the advantage of pre-training needs justification for some datasets.


[1] Advanced graph and sequence neural networks for molecular property prediction and drug discovery
Bioinformatics 38 (9), 2579-2586, 2022

[2] MoleculeNet: a benchmark for molecular machine learning
Chem Sci. 2018 Jan 14; 9(2): 513–530.

---

> ### Author Response · Authors · 2023-03-20
> **Response to Reviewer k4tC**
>
> Thank you for providing review for improving the clarity and the quality of our work. We have the following response for your questions.
>
> **Q1: The novelty is limited. There are recent works that also leverage multiple featurizations [1].**
>
> Thanks for your valuable comment. Please refer to our general response regarding the novelty and the technical contribution of this work.
>
> **Q2: The reported numbers, e.g. Table 2, are lower than simple supervised baselines reported in [2]. And some metrics do not follow the recommendations in [2] (PRC-AUC vs ROC-AUC). In addition, the advantage of pre-training needs justification for some datasets.**
>
> Our experiments were primarily conducted based on earlier work [3] and widely used settings in related works. Compared to MoleculeNet [2], there are three main differences in our experimental settings that may account for the lower numbers:
>
> 1. Different number of features used: In the settings we used, only atom number and chirality tag are used as node features, which is less than the seven features used in MoleculeNet.
> 2. Different dataset splitting methods: We used scaffold split for all downstream datasets, while MoleculeNet only used this splitting method for three datasets (HIV, BBBP, and BACE).
> 3. Different prediction heads: In our experiments involving multiple property prediction targets, the output embeddings correspond to task dimensions, while MoleculeNet classifies each task separately and takes the average for prediction.
> Such reasons render that our reported performance not directly comparable to MoleculeNet.
>
> [3] Strategies for Pre-training Graph Neural Networks Hu et.al. ICLR 2020

---

### Author Response · Authors · 2023-03-20
**General Response: thank you & response to common concerns**

We would like to thank all reviewers for their extensive reviews and constructive critiques. We are encouraged that reviewers find that our approach is simple and effective (Reviewer 2Ygj and P88d), that the ablation study is sufficient (Reviewer P88d), that the motivation is solid (Reviewer k4tC), and that the learned molecular representation demonstrates good empirical performance on downstream tasks (Reviewers 5QhU and P88d).

In the following, we would like to make clarifications regarding concerns on the novelty and technical contributions raised by reviewers:

**Q: Novelty and technical contributions.**

**A**: The novelty and technical contributions of this work can be summarized from the following three aspects.
Firstly, we for the first time comprehensively utilize four widely used featurization techniques for molecular representations, covering string-, graph-, scalar-, vector-based representations for 1D/2D molecules and 3D conformation structures. We also adopt proper encoders for every featurization. Among them, since there is a lack of proper neural encoders for fingerprints, we propose an attention-based network to model interactions of feature fields in fingerprints, honoring the discrete and extremely sparse nature of fingerprint vectors. Despite the fact that other encoders are taken from existing literature, they are representative and frequently used in the community, which enables direct and fair comparison with respect to prior work. We would like to note that our proposed MOCO framework is generic and could be applied to any off-the-shelf encoders and molecular pretraining tasks.

Secondly, we propose a novel molecular pretraining framework that adaptively integrates information from multiple views in both pretraining and fine-tuning stages. This pretraining mechanism is different from all previous work (e.g., GraphCL, 3D Infomax, and GraphMVP) that only considers 2D graph structures during fine-tuning. Through extensive experiments, we empirically demonstrate that our adaptive multiview fusion involving both 1D/2D representations and 3D conformations are helpful for a variety of molecular property prediction tasks. Note that previous work has already achieved pretty high performance. For example, the current state-of-the-art GraphMAE only obtains a 1.6% absolute improvement over its best baseline in terms of average ROC-AUC. Our work pushes that boundary without extensive hyperparameter tuning, achieving a 1.1% absolute improvement over GraphMAE and 2.72% absolute improvement over GraphMVP in terms of average ROC-AUC.

Thirdly, we would like to clarify that we aim to rethink whether commonly used featurizations (e.g., 2D graph and 3D geometry) together with appropriate neural networks (e.g., GIN and SchNet) properly depict the biology and chemistry space for molecular property prediction tasks. Inspired by two simple case studies, we find that the featurizations differ in their ability to recognize different semantics. Thus, our work reveals the significance and necessity to incorporate other featurizations and corresponding encoders to complement the unnoticed information, which also leaves room for the community to reconsider the importance of different molecular modalities.

Once again, we appreciate all reviewers for their insightful suggestions which enable us to further strengthen our work. We expect that our response will resolve the reviewers' concerns. We are happy to address any further concerns or comments that reviewers may have.

---

### Author Response · Authors · 2023-04-16
**Any further concerns or comments?**

Dear reviewers,

Thanks again for your valuable reviews! We are committed to addressing any further concerns you may have.

Sincerely,
Authors

---

### Decision · Action_Editors · 2023-05-01

**Recommendation:** Reject

**Comment:**

This paper proposed a framework for molecule pretraining. During the initial review period, reviewers raised some concerns about this paper, some of which were later addressed by the authors' rebuttal. Reviewers believe that this paper's novelty is somewhat limited, and there are more questions that need to be discussed.  Specifically,

Reviewer 5QhU expressed that one of his concerns was not fully addressed in the rebuttal.  The paper claims that their proposed model is better than state-of-the-art models, but there are better-performing models cited in other papers, such as GraphMVP-G or GraphMVP-C. The response suggests that these SSL strategies can be applied to any model, but it is not clear why the authors did not show the results of applying those techniques to their model.

In addition, Reviewer k4tC has pointed out two major concerns that haven't been addressed. I quoted Reviewer k4tC's concerns below:

1. The authors explained the lower numbers in experiments by clarifying that they were using fewer features than MoleculeNet [2]. This directly leads to the question: How do the missing features and other molecule featurizations contribute to the performances, respectively? It is likely that, with more features, the gain from other molecule featurizations is saturated.

2. The authors also mentioned that different prediction heads could account for lower numbers. In my understanding, the reason behind this difference is because of the pre-training fine-tuning design. As I mentioned in my review, the advantage of pre-training needs justification for some datasets. The fact that different prediction heads could account for lower numbers is showing the disadvantage instead.

The experimental results are less convincing to the Reviewer k4tC.

Given these remaining unsolved questions and after discussion with the editor, we believe it would be beneficial for the authors to address these major concerns and submit a major revision.


**Audience:**

Yes.

**Claims And Evidence:**

Reviewers feel that a part of the claim made in this paper is not well-supported by the experimental results. Reviewer 5QhU observed that the proposed algorithms were not applied to models with state-of-the-art performances, such as GraphMVP-G or GraphMVP-C. This lack of application to high-performing models weakens the claim of model agnosticism and outstanding performance, making it less convincing. Reviewer k4tC also expressed concerns regarding the experiment results, specifically the comparison with MoleculeNet. The unconvincing weak results compared with baselines lead Reviewer k4tC to believe that the main claim of this paper is inadequately supported.